# Mating-Induced Trade-Offs upon Egg Production versus Fertilization and Offspring’s Survival in a Sawfly with Facultative Parthenogenesis

**DOI:** 10.3390/insects12080693

**Published:** 2021-08-02

**Authors:** Hong Yu, Min-Rui Shi, Jin Xu, Peng Chen, Jian-Hong Liu

**Affiliations:** 1Yunnan Academy of Biodiversity, Southwest Forestry University, Kunming 650224, China; yuhong1652@126.com (H.Y.); smr1997by@163.com (M.-R.S.); 2Key Laboratory for Forest Resources Conservation and Utilization in the Southwest Mountains of China, Ministry of Education, Southwest Forestry University, Kunming 650224, China; 3Yunnan Academy of Forestry and Grassland, Kunming 650201, China; pengchenn@126.com

**Keywords:** *Cephalcia chuxiongica*, mating-responsive genes, egg production, egg fertilization, offspring’s survival, somatic maintenance

## Abstract

**Simple Summary:**

Study of mating-induced trade-offs between reproduction and survival is conducive to provide evolutionary insights into reproductive strategies and aging. Using RNA sequencing and bioinformatics, we found that mating induced changes of genes and pathways related to reproduction and survival in females of a pine sawfly. Mating induced substantial downregulation on genes associated to immunity, stress response, and longevity. However, mating induced divergent reproductive response, with downregulation on genes related to egg production while upregulation on genes related to egg fertilization. Considering the nature of limited resources in adults, low fecundity and egg protection behavior in this sawfly, we suggest that mating triggers trade-offs between reproduction and survival in this insect and females of this species have evolved specific strategies to adapt to the living conditions, e.g., restrict whole fecundity to ensure higher fertilization and offspring’s survival.

**Abstract:**

Investigation of mating-induced trade-offs between reproduction and survival is conducive to provide evolutionary insights into reproductive strategies and aging. Here, we used RNAseq and bioinformatics to reveal mating-induced changes of genes and pathways related to reproduction and survival in female *Cephalcia chuxiongica*, a pine defoliator with facultative parthenogenesis and long larval dormancy. Results showed that mating induced substantial downregulation on genes and pathways associated to immunity, stress response, and longevity. However, mating induced divergent reproductive response, with downregulation on genes and pathways related to egg production while upregulation on genes and pathways related to egg fertilization. Considering the nature of limited resources in adults, low fecundity, and egg protection behavior in *C. chuxiongica*, we suggest that mating triggers trade-offs between reproduction and survival in this insect and females of this species may have evolved specific strategies to adapt to the environmental and hosts’ conditions, e.g., restrict whole fecundity to ensure higher fertilization and offspring’s survival. Moreover, mating induced significant responses on genes and pathways that play important roles in vertebrate reproduction while their function in insects are unclear, such as the progesterone-mediated oocyte maturation pathway; the significant regulation after mating suggests that their function may be evolutionarily conserved in animal kingdom.

## 1. Introduction 

Trade-offs between reproduction and survivorship is the central theme in evolutionary biology of senescence. Negative relationships between reproduction and survivorship are often observed in organisms, in which increased fecundity may cause costs on somatic maintenance (such as immunity and stress responses) and longevity [1,2,3,4]. The resources allocation model suggests that the trade-offs between reproduction and survival is likely due to alternative allocation of limiting energetic resources because both reproduction and soma maintenance are energetically costly [1,5]. Genetic model suggests that there are antagonistic alleles that may promote negative genetic correlation between reproduction and survival, i.e., alleles may promote fecundity at the expense of somatic maintenance [2,6]. These two models are not mutually exclusive [2,7].

In insects, the endocrine network plays vital roles in reproduction and maintenance regulation, which mainly involves insulin-like/IGF-1 signaling (IIS), juvenile hormone (JH), 20-Hydroxyecdysone (20E), yolk precursor vitellogenin (Vg), and yolk proteins (YPs) [2]. JH and IIS are important mediators of egg maturation; improved JH and IIS levels promote the expression of Vg and YPs and the uptaking of Vg and YPs into oocytes [1]. However, elevated JH and IIS may inhibit immune responses, such as by reducing phenoloxidase activity and the production of antimicrobial peptides [1,8]. In contrast, 20E may promote immune responses, such as by upregulating the expression of antimicrobial peptide genes in *Drosophila melanogaster* [8,9]. Therefore, although the mechanism remains poorly understood, such opposite effects of different hormones seem to be intimately involved in the trade-offs between reproduction and survivorship [1,2].

Female insects are likely to face a limited protein supply during reproductive period because many of them do not feed on a protein source as adults while elevated egg production process requires higher protein supplies at this stage [10,11,12]. Female insects thus may have evolved behavioral and physiological strategies to allocate limited resources optimally between reproduction and survival [1,13]. Mating is a key switch for sexual reproduction in insects, which incurs major changes in the physiology and behavior in females [14]. Previous studies have found that mating did positively affect female reproduction and negatively affect female immune activities in some insects [1,5,15]. However, some studies also found that there is no difference in immune responses between mated and virgin insect females and in some insects mating even upregulates female immune responses [5]. These diverse findings may be due to the differences in measurement methods and immune indicators and the differences in species and mating systems. For example, males may directly suppress female immunity, thereby promoting sperm storage and egg fertilization by their sperms in female reproductive tracts [16]. However, mated females may need to have higher post-mating immune activity to provide defense against mating transferred infections, particularly in polygamous mating systems [5,17]. The heat shock response is essential for proteostasis and cellular health [4]. Similarly, studies on reproduction-induced heat shock response also found inconsistent results. In *C. elegans*, the heat shock response was inhibited at the onset of reproduction, probably due to competing requirements of the germline and soma [4]. In honey bees, however, increased reproduction did not cause loss of heat shock response in the reproductive queen [3]. Therefore, honey bee queens may possess an atypical uncoupling of the reproduction-maintenance trade-off, in which stress response may be maintained during reproduction [3]. Therefore, study on the trade-offs between reproduction and survival in different mating systems by using methods that can more comprehensively examine various indicators (such as high-throughput sequencing and bioinformatics) will provide deeper insights in this field.

Recent progress on high-throughput sequencing and bioinformatics has revolutionized our understanding of life and organism. Previous transcriptome analysis in a number of insect species have shown that mating can induce expression changes in genes related to reproduction, immunity, stress response, and longevity [18,19,20]. A resent transcriptome-based study in the sweet potato whitefly females found 434 deferentially expressed genes (DEGs) between mated and unmated groups, with many of them encoding binding-related proteins and genes associated with lifespan [20]. Another study in *Spodoptera litura* female moths showed a divergent response in DEGs in relation to reproduction and immunity [15]. These results suggest that trade-offs on reproduction versus survival were induced by mating in some insects. In the present study, we studied genes and pathways related to reproduction and survival (immunity, stress response, and longevity) and mating-induced regulation in females of a pine sawfly, *Cephalcia chuxiongica*. This insect has a facultative parthenogenesis reproductive system and a long (19 months) larval dormancy stage [21]. Study of mating-induced regulation on reproduction and survivorship in such a special insect species is expected to achieve some new enlightenment on the evolution of reproductive strategies and aging.

The sawfly *Cephalcia chuxiongica* Xiao (Hymenoptera: Pamphiliidae) is a serious pest of pines in China [22,23]. Almost fifty *Cephalcia* species have been found all over the world, with most of them being important forest pests [24,25]. *Cephalcia chuxiongica* larvae feed on pine needles and pupate in the soil under the trees. The life cycle of this species is long (about 22 months) due to a 19-month larval diapause in the soil [22]. Similar to many other species from Hymenoptera, *C. chuxiongica* females perform sexual reproduction under normal conditions, while in the absence of males, they can perform parthenogenesis [22]. The male and female adults of *C. chuxiongica* can mate a few hours after eclosion; unmated females (virgin) may start to lay unfertilized eggs through parthenogenesis a few days (>3 d) after eclosion [21]. Sawflies do most of their feeding during the larval stage, and adults of some species may feed on pollen and nectar [26]. So far, there is no report on adult feeding habits in *C. chuxiongica*; adults even did not feed on the provided honey solution, suggesting that this species do not feed during the adult stage [21].

## 2. Materials and Methods

### 2.1. Sample Collection

*Cephalcia chuxiongica* adults were collected in August 2019 under infested pine trees in the *Pinus yunnanensis* forest, located on a small mountain (25.600166N, 103.439364E) near Xundian town in Yunnan Province, China. *Cephalcia chuxiongica* pupate and eclose underground during July to Sept. Eclosed adults crawl out from the soil for reproduction in the forest. To confirm adults virginity and age, newly eclosed adults were directly dug out from the soil below the pine trees in the morning [27]. Males and females were sexed according to morphological traits [23] and reared separately on the pine trees (on the pine needles covered by nylon mesh bag [30 cm diameter, 40 cm long] to avoid insect escape). As mentioned above, *C. chuxiongica* adults can mate soon after emergence and mating mostly happen at noon. Therefore, mating was allowed in the same day after collection during 12:00–14:00 by pairing males and females on the pine needles covered by bags with one pair per bag. Mating events (two insects engaged at the tip of the abdomen; mating duration is about 20 min) [22] were recorded. Males were removed after mating and the mated females were then individually reared in the same bag and their abdomens were sampled at 1 h, 6 h, and 24 h after mating. Abdomens from fifteen females were used as a replicate, and three replicates were used for each sampling time point. Virgin females at the same age as mated individuals were used as controls. All samples were placed in liquid nitrogen for immediate freezing after sampling and stored at −80 °C. One replicate from Virgin-1 h and one replicate from Virgin-6 h were discarded due to lower sequencing quality (see below) and thus finally, two replicates for Virgin-1 h and Virgin-6 h, and three replicates for other treatments (Virgin-24 h, Mated-1 h, Mated-6 h and Mated-24 h) were used for analysis.

### 2.2. Library Preparation and Sequencing

Total RNA was extracted from samples using Trizol reagent (Invitrogen Inc., Calsbad, CA, USA) and then was treated with the RNase-free DNase I to eliminate genomic DNA. The purity and concentration of RNA was assessed by using Qubit RNA Assay Kit (Life Technologies Inc., Grand Island, NY, USA) and the NanoPhotemeter spectrophotometer (Implen Inc., Westlake Village, CA, USA). The RNA integrity was checked by the Agilent Bioanalyzer 2100 system (Agilent Technologies Inc., Palo Alto, CA, USA). One microgram RNA per sample was used for the preparation of the sequencing libraries by using NEBnext Ultra RNA Library Prep Kit for Illumina (New England BioLabs Inc., San Diego, MA, USA) following the manufacturer’s instructions and index codes were added to attribute sequences to each sample. The quality of each sample library was assessed using the Agilent Bioanalyzer 2100 system. Ultimately the library per sample was sequenced by Illumina Hiseq4000 platform (Majorbio Biotechnology Inc., Shanghai, China) and the paired-end reads were generated.

### 2.3. Quality Control and Assembly

The raw reads obtained were initially processed by trimming the adapter and low quality reads to produce clean reads. The clean reads were assembled using the Trinity software (version 2.5.1) to generate transcripts [28]. Then transcript analysis was performed to remove redundancies with TGICL software (version 2.1) and acquire unigenes without redundancy [29]. 

### 2.4. Differential Expression Analysis

Gene expression levels were determined as transcripts per million (TPM). The differential expression analysis between samples was performed using the edgeR R package (3.0.8). *p*-value was adjusted using *q*-value [30]. *q* < 0.05 and |log2(foldchange)| > 1 was set as the threshold for significantly differential expression.

### 2.5. Functional Annotation and Enrichment Analysis of Differentially Expressed Genes (DEGs)

Using the BLAST [31] software, the DEGs were compared with NCBI non-redundant protein (NR), Swiss-Prot protein database (Swiss-Prot), Gene Ontology (GO), KEGG Orthology (KO), Cluster of Orthologous Groups (COG), and Pfam databases to obtain annotation information about the DEGs. GO enrichment analysis of DEGs was implemented by using the GOSeq program and KEGG enrichment was performed using the KOBAS software. GO terms and KEGG pathways with *q* < 0.05 were significantly enriched in DEGs.

### 2.6. Validation by qRT-PCR

To verify the accuracy of the RNAseq data, 24 DEGs were used for qRT-PCR and the *β-Tubulin* (GeneBank ID: MZ028603) was used as a reference gene. Total RNA was extracted from above samples by using RNAiso plus (TaKaRa Inc., Dalian, China) and cDNA was synthesized using PrimeScript RT reagent Kit (Takara Inc., Dalian, China). The PCR was performed by QuantStudio 7 Flex (Thermo Fisher Scientific Inc., Waltham, MA, USA) with gene specific primers (Appendix A) and the following program: 95 °C for 30 s, followed by 40 cycles of 95 °C for 5 s, 60 °C for 30 s, and dissociation. The 2^−ΔΔCT^ method [32] was used to calculate the relative expression. Differences of gene expression levels between treatments were analyzed by ANOVA using SPSS 25.0. The rejection level was set at *α* < 0.05.

## 3. Results

### 3.1. Sequencing and Assembly

By RNAseq using Illumina HiSeq4000 platform, ~55,000,000 clean reads were obtained from each of the 16 sequenced libraries (Appendix A). The percentages of Q20 and Q30 of all samples’ clean reads ranged from 97.76% to 98.21% and from 94.10% to 95.21%, respectively. The biological replicates were highly correlated (Appendix A), which affirmed the reproducibility of the RNASeq technology and biological replicates. The transcriptome raw reads were deposited to the NCBI SRA database (accession no.: SRR13991392~SRR13991407).

### 3.2. Outline of Mating-Induced Transcriptional Changes

There are 149, 320, and 1068 DEGs within Mated-1 h vs. Virgin-1 h (M1 h-vs.-V1 h), M6 h-vs.-V6 h and M24 h-vs.-V24 h groups, respectively (Figure 1; Appendix A). There were 89 DEGs shared between M1 h-vs.-V1 h and M1 h-vs.-V1 h, 4 DEGs shared between M1 h-vs.-V1 h and M24 h-vs.-V24 h, and 36 DEGs shared between M6 h-vs.-V6 h and M24 h-vs.-V24 h, while only one DEG (*DN61334_c1_g3*, *hemolymph glycoprotein*, downregulated) was shared by the three groups (M1 h-vs.-V1 h, M6 h-vs.-V6 h and M24 h-vs.-V24 h) (Figure 2; Appendix A).

To better understand their functions, all these DEGs were annotated based on NR, Swiss-Prot, GO, KO, COG, and Pfam databases (Appendix A) and then were submitted for GO (Appendix A) and KEGG (Appendix A) enrichment analysis. Most of these DEGs (1235/1537 = 80.4%) were successfully mapped to at least one of these databases (Appendix A). A total of 309 GO terms were significantly enriched in these DEGs, with most of them (59.5%) being assigned to biological process (BP) terms, and then 17.2% and 23.3% being assigned to cellular component (CC) terms and molecular function (MF) terms, respectively (Figure 3a and Figure 4; Appendix A). Moreover, there are 51 KEGG pathways that were significantly enriched in these DEGs, which were assigned to six of Level I Categories (Figure 3b and Figure 5; Appendix A).

Mating-induced regulation on genes and pathways related to reproduction and survivorship were then studied in detail as follows based on the above annotation and enrichment analysis.

### 3.3. Transcriptional Changes at 1 h Post-Mating

At 1 h after mating, 68 genes were downregulated and 81 genes were upregulated in mated females compared to virgin females (Figure 1a). The log2FoldChange (LC) value of DEGs varied from −13.04 to 12.74 (Appendix A). 

Two reproduction-related genes were found within these DEGs, with one encoding follicle cell protein (downregulated) and one encoding cytochrome P450 (upregulated) (Table 1). KEGG enrichment analysis of the 149 DEGs found one reproductive related pathway, the steroid hormone biosynthesis pathway, which enriched to one upregulated DEG (Table 2).

Two immunity related genes were found within these DEGs (Table 3). Both of them encode antimicrobial peptides and both were downregulated in mated females compared to virgin ones. Six immunity related pathways were enriched based on the 149 DEGs, with four pathways enriched to downregulated DEGs and two pathways enriched to upregulated DEGs (Table 4).

No longevity and heat shock-related genes and pathways were found within these DEGs of M1 h-vs.-V1 h.

### 3.4. Transcriptional Changes at 6 h Post-Mating

Within the 320 DEGs of M6 h-vs.-V6 h, 206 were downregulated and 114 were upregulated in mated females compared to virgin ones (Figure 1b). The LC value of DEGs differed from -10.34 to 12.29 (Appendix A).

Sixteen reproductive-related genes (all downregulated) were found within these DEGs (Table 1), including five Vg encoding genes (LC: −9.15 to −2.35), three Haemolymph juvenile hormone-binding protein-encoding genes (LC: −3.45 to −2.48), two Takeout-like protein encoding genes (LC: −2.34 to −1.59), one UDP-glucosyltransferase like encoding gene (LC: −2.71), one matrix metalloproteinase encoding gene (LC: −8.16), and four cytochrome P450 encoding genes (LC: −2.86 to −2.01). No reproductive-related KEGG pathways were significantly enriched to the 320 DEGs.

One immunity-related gene was found within these DEGs (Table 3), which is the phenoloxidase-encoding gene (LC: −3.53). Three immunity-related KEGG pathways were significantly enriched to 12 downregulated DEGs (Table 4).

Still no longevity and heat shock related genes and pathways were found within these DEGs.

### 3.5. Transcriptional Changes at 24 h Post-Mating

Within the 1068 DEGs of M24 h-vs.-V24 h, 104 genes were downregulated and 964 genes were upregulated in mated females compared to virgin ones (Figure 1c). The LC value of DEGs changed from −10.50 to 14.64 (Appendix A).

A total of 51 transcripts were annotated as reproductive related genes within these DEGs (Table 1), including seven *Vitellogenin* and *Vitellogenin-like* (all downregulated), one *Vitellogenin-receptor* (upregulated), 26 *Zona pellucida protein-like* (all upregulated), seven *Cyclin-like* (all upregulated), and ten other reproductive-related genes. Three reproductive related KEGG pathways were significantly enriched (Table 2): (1) Estrogen signaling pathway, enriched to four downregulated DEGs; (2) oocyte meiosis pathway, enriched to 18 upregulated DEGs; and (3) progesterone-mediated oocyte maturation pathway, enriched to 13 upregulated DEGs. To get some evolutional clues on oocyte maturation, unigenes annotated to the progesterone-mediated oocyte maturation pathway (Appendix A) were mapped to the same pathway of *Xenopus* (KEGG id: map04914) (Figure 6). Results show about half of the genes on map04914 were found in the unigenes of *C. chuxiongica*. This study also revealed an additional 25 unmapped progesterone-mediated oocyte maturation-related unigenes (Appendix A).

Three immunity related genes were found within these DEGs with all of them being downregulated (Table 3). Thirteen immunity-related KEGG pathways were significantly enriched, with ten of them being enriched to downregulated DEGs and three being enriched to upregulated DEGs (Table 4). 

There are thirteen Hsps-encoding genes found within these DEGs (Table 3), including four sHsps (LC: −6.63 to −3.69), seven Hsp70s (LC: −6.23 to 10.90), and two Hsp90s (LC: 3.93 to 7.39).

One longevity-related pathway was significantly enriched, which was related to four downregulated DEGs (Table 4). 

### 3.6. Validation of RNAseq by qRT-PCR

Eight DEGs from each group (M1 h-vs.-V1 h, M6 h-vs.-V6 h and M24 h-vs.-V24 h, respectively) were used to verify the accuracy of RNAseq, which include six reproduction-related genes (*DN64818_c0_g6*, *DN52313_c1_g5*, *DN65191_c2_g1*, *DN66978_c2_g1*, *DN78694_c0_g1*, and *DN71364_c1_g1*), three immunity-related genes (*DN70573_c2_g1*, *DN60391_c5_g2* and *DN63597_c1_g1*), and one heat shock response gene (*DN70765_c7_g9*) (Figure 7). The expression levels of these genes measured by qRT-PCR were similar to the results of RNAseq analysis, which suggested that the RNAseq data were reliable.

## 4. Discussion

Previous studies generally found that mating induces the upregulation of egg production (fecundity)-related genes, such as YPs and Vg [15,18,19]. In the present study, however, we found an opposite result, where all Vg encoding DEGs (five in M6 h-vs.-V6 h and seven in M24 h-vs.-V24 h) were found downregulated largely (Table 1). Moreover, other positive modulator of fecundity, including *Follicle cell protein, Haemolymph juvenile hormone binding protein*, *UDP-glucosyltransferase*, *Apolipophorin III*, and *Matrix metalloproteinase*, were also downregulated significantly (Table 1). In addition, the upregulation of *Insulinase* (which destroys or inactivates insulin) in mated females may also negatively affect female fecundity. These results suggested that mating downregulated reproduction in terms of fecundity (number of eggs) in *C. chuxiongica*.

On the contrary, we found a significant upregulation on genes related to oocyte maturation and embryogenesis (Table 1): (1) Seven *Cyclins*, which play roles in the process of oocyte maturation and the onset of embryogenesis [47,48,49]; (2) two *Zygote arrest protein*, which are ovary-specific maternal factors that play essential roles during the oocyte-to-embryo transition [51]. Moreover, two oocyte maturation-related pathways, the oocyte meiosis pathway and the progesterone-mediated oocyte maturation pathway, have been significantly enriched upregulated DEGs (Table 2). Fully grown oocytes are arrested in the first meiotic prophase (Figure 6), which is a state of low metabolic activity without detrimental effects on subsequent embryogenesis [57]. At the end step of oocyte maturation, a process termed meiotic reinitiation or resumption, takes place before or during the time when the oocyte moves from the follicle to the oviduct (ovulation). Meiotic maturation involves the activation of various signal transduction pathways, which converge to activate maturation-promoting factors (Figure 6). In vertebrate, the marks of meiotic maturation include [58]: (1) Resumption of meiosis I, including germinal vesicle breakdown, chromosome condensation, and spindle formation; (2) the transition from meiosis I to meiosis II; and (3) arrest in metaphase II. Meiosis II will complete after egg fertilization. In many invertebrates, however, oocytes maturation proceeds only to metaphase of meiosis I, which is when they are fertilized [58]. Therefore, the upregulation of genes and pathways in relation to oocyte meiosis and maturation in *C. chuxiongica* (Table 1 and Table 2) are likely to function in meiotic resumption and maturation in fully grown oocytes, from which the egg is ready for the following fertilization. 

The progesterone-mediated oocyte maturation pathway of the African clawed frog *Xenopus laevis* is the most intensively studied model system for meiotic maturation [58]. Therefore, we mapped unigenes that annotated to the progesterone-mediated oocyte maturation pathway to the same pathway of *Xenopus* (KEGG id: map04914) and found that about half of the genes on map04914 were present in the unigenes of *C. chuxiongica* (Figure 6; Appendix A). Further searching found an additional 25 unmapped progesterone-mediated oocyte maturation-related unigenes in *C. chuxiongica* (Appendix A). These results suggest that this pathway may also play roles in egg maturation of *C. chuxiongica* but factors and signaling pathways may be species-specific. In the present study, we also found the presence of Cyclin-B1 (KO id: K05868) and Xkid (KO id: K10403; kinesin-like DNA-binding protein, also known as Kinesin-like protein, KIF22) [57] in the enriched DEGs (Figure 6, Appendix A). These proteins are all dispensable for meiosis I entry but play essential roles in the progression from meiosis I to meiosis II [58]. As mentioned above, oocytes maturation in many invertebrates proceeds only to metaphase of meiosis I until they are fertilized. If it is the case in *C. chuxiongica*, then the upregulation of Cyclin-B1 and Xkid may be triggered by mating factors (such as male accessary protein and sperms) or fertilization, which then will promote the transition of eggs from meiosis I to meiosis II. This warrants further studies.

More interestingly, this study also found significant expression changes in a number of genes that may relate to fertilization and egg hatching, including a number of *Zona pellucida protein* and *Ovastacin*/*Astacin-like* (Table 1). The Zona pellucida (ZP) is a glycoprotein layer surrounding the plasma membrane of mammalian oocytes, which is a vital constitutive part of the oocyte and functions in primary binding and induction of the sperm acrosome reaction [42,44]. A family of 18 ZP-like protein encoding genes has been identified in *D. melanogaster* [59]. These genes are specifically expressed during embryogenesis and during differentiation of epithelial tissues. However, whether these ZP-like proteins also play roles in fertilization is still unknown in *D. melanogaster* or other insect species because little is known about the molecular mechanism of fertilization in insects [59]. Ovastacin plays a role in the post-fertilization block to sperm binding by cleaving ZP2 in the ZP to prevent polyspermy in mouse [52]. Some astacins may function as hatching enzyme in insects, which degrades the egg envelope to release the embryo [60]. The reproductive-related regulation on these genes may suggest they are crucial in the reproduction process of insects.

Insect cytochrome P450 families contain a class of enzymes, which play diverse functions in detoxification and the biosynthesis of hormones [34]. In *Drosophila*, mating induced downregulation in six P450 genes and upregulation in 22 P450 genes. In *B. tabaci*, mating induced upregulation in two P450 genes [20]. In the present study, one P450 gene (*CYP2C*) was significantly upregulated in mated females shortly after mating (1 h post-mating) and four P450 genes were downregulated in mated females at 6 h post-mating (Table 1). Upregulation of P450 shortly after mating may be a detoxification response as males may transfer slightly toxic seminal fluid during mating [61] and downregulation of P450 sometime after mating may be due to trade-offs between reproduction and survival.

Reproduction associated heat shock responses are inconsistent in different species, which is downregulated after the onset of reproduction in *C. elegans* [4] while in honey bees, increased reproduction did not cause loss of heat shock response in the reproductive queen [3]. In *B. tabaci*, mating led to the upregulation of Hsp68 and Hsp70 genes at different time points in mated females [20]. In the present study, we also found significant expression changes in thirteen Hsps-encoding transcripts 24 h post-mating, with some of them (four sHsps and four Hsp70s) being downregulated and others (three Hsp70s and two Hsp90s) being upregulated substantially (Table 3). Hsps act as molecular chaperones to improve organisms’ survival, development, and reproduction under different stresses. Studies have shown that Hsps have multiple functions during reproductive process, including gamete protection [58], oocyte maturation (Figure 6) [62], reproduction vs. survival trade-offs [63], etc. In addition, one longevity-associated pathway was significantly enriched, which related to four downregulated Hsp70s coding DEGs (Table 4). Therefore, mating induced different response in different Hsps may be due to their different functions in reproduction and survival.

Insect immune defenses are carried out through humoral and hemocyte responses, with the former playing defensive role through the synthesis of antimicrobial peptides and the latter through encapsulation and phagocytosis [64]. In the present study, four antimicrobial peptides encoding gene (*Defensin*, *Hymenoptaecin*, *Megourin*, and *Fungal protease inhibitor*) were found in *C. chuxiongica* and all of them were downregulated considerably in mated females (Table 3). In addition, two hemocyte responses related enzyme-encoding genes, *Phenoloxidase* and *Lysozyme*, were also found downregulated in mated females (Table 3). Lysozyme is involved in immune defense by hydrolyzing the bacterial cell walls whereas phenoloxidase participates in the synthesis of oxidative free radicals and the defensive melanization [1]. A total of 21 immunity related KEGG pathways were enriched, with sixteen of them being enriched to downregulated DEGs and five being enriched to upregulated DEGs. These results have suggested that the immunity activity has been downregulated substantially in mated females.

Above results and discussion have suggested that mating is also an essential switch in *C. chuxiongica*, a species with facultative parthenogenesis, which induced significant trade-offs between reproduction and survivorship. As mentioned above, the male and female adults of *C. chuxiongica* can mate a few hours after eclosion and mated females start to lay fertilized eggs in the subsequent day after mating, as well as start to show egg protection behavior; unmated females (virgin) may start to lay unfertilized eggs through parthenogenesis a few days (>3 d) after eclosion [21]. Therefore, the pattern of gene expression changes after mating is consistent with the post-mating behavioral and physiological changes. Moreover, the present study also suggested that mating may also induce trade-offs within reproduction on fecundity vs. fertility in *C. chuxiongica* females, in which females may downregulate oogenesis and egg production to restrict fecundity but upregulate meiotic maturation in fully grown oocytes and fertilization-related pathways to favor the following egg fertilization. *C. chuxiongica* mostly occurs in pine forests at high altitude and barren soil, its larvae are obligate pine defoliator, which mainly feed on *Pinus yunnanensis* [21]. *C. chuxiongica* has a long (19 months) larval dormancy stage and adults do not feed [21], which may force adults to use limited resources optimally on reproduction and survival. In addition, *C. chuxiongica* females lay eggs on the surface of pine needles [23], which will facilitate larval feeding after hatching but also will likely to incur attack by natural enemies. *C. chuxiongica* females may thus have evolved egg protection behavior. The fecundity of *C. chuxiongica* is about 50 eggs per female [22]. Therefore, compared to other high fecundity insect species, such as *S. litura* that lay more than one thousand eggs per female [65] and show upregulation of fecundity related genes after mating [15], *C. chuxiongica* may have evolved a different reproductive strategy, i.e., restrict whole fecundity while ensuring higher egg fertilization and offspring survival. Future studies to clarify the hypotheses established in this study by using other techniques, such as RNAi and 2D electrophoresis/MS, will help to provide deeper insights in this field.

The present study and previous studies [21,22,23] on *C. chuxiongica* suggest that this species has evolved multiple physiological and behavioral strategies (such as facultative parthenogenesis, long larval diapause, and trade-offs on reproduction and survival) to adapt to its living environment.

## Figures and Tables

**Figure 1 insects-12-00693-f001:**
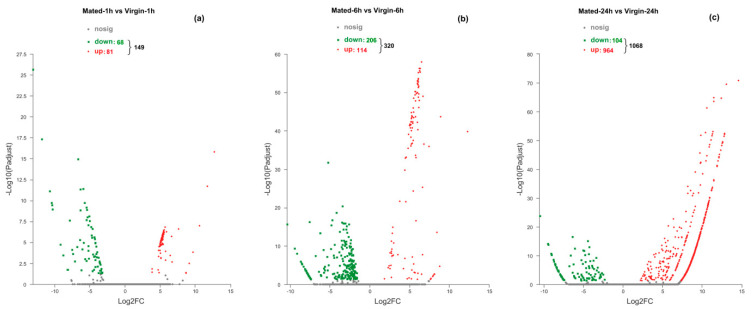
Volcano plots of the DEGs in Mated-1 h vs. Virgin-1 h group (**a**), Mated-6 h vs. Virgin-6 h group (**b**), and Mated-24 h vs. Virgin-24 h group (**c**). Genes with significant differential expression were indicated by red dots (upregulated) and green dots (downregulated). Genes with no significant differential expression were represented by grey dots.

**Figure 2 insects-12-00693-f002:**
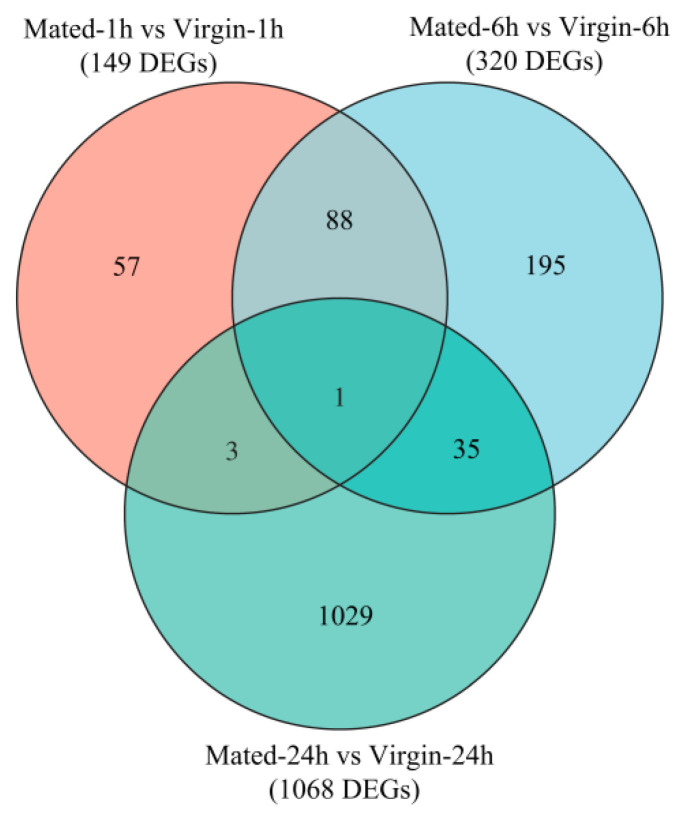
The Venn diagram of DEGs. The overlapping circles represented common DEGs among all combinations.

**Figure 3 insects-12-00693-f003:**
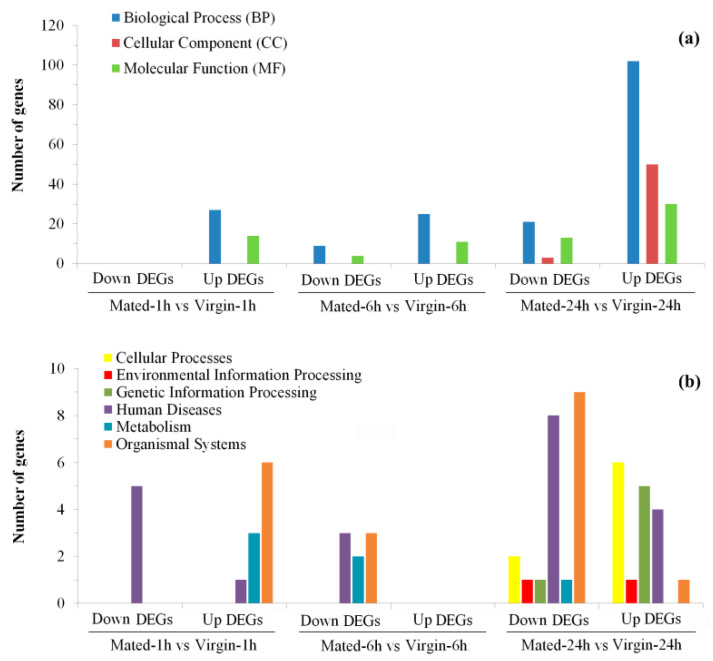
Summary of GO (**a**) and KEGG (**b**) enrichment analysis of DEGs in different groups.

**Figure 4 insects-12-00693-f004:**
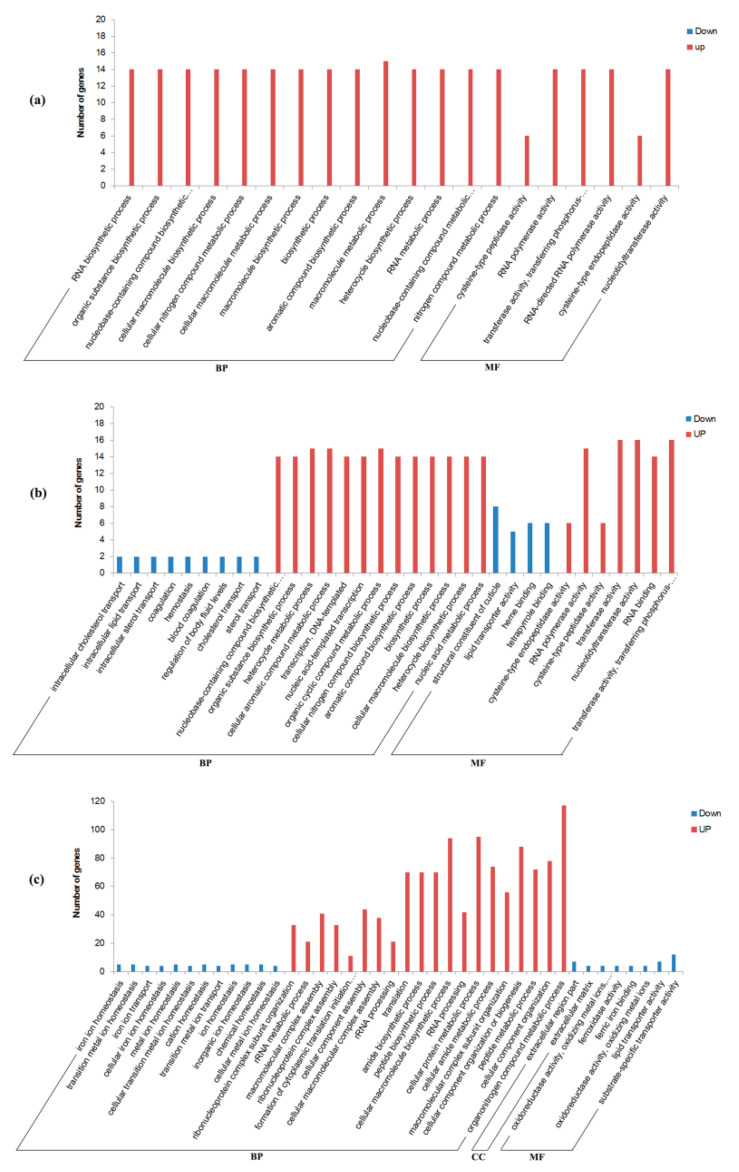
GO enrichment of DEGs in Mated-1 h vs. Virgin-1 h group (**a**), Mated-6 h vs. Virgin-6 h group (**b**), and Mated-24 h vs. Virgin-24 h group (**c**). The function of DEGs was divided into three parts: BP (biological process), CC (cell composition), and MF (molecular function). The red bars indicate upregulated DEGs and blue bars indicate downregulated DEGs. The top 20 terms each from upregulated DEGs and downregulated DEGs of each group are shown.

**Figure 5 insects-12-00693-f005:**
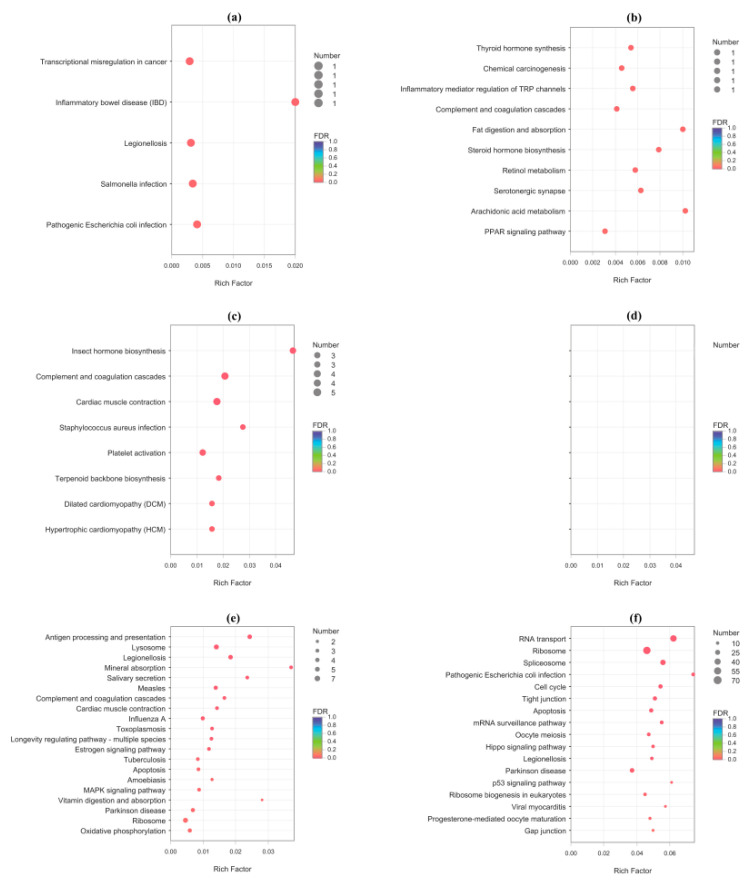
KEGG pathway enrichment of DEGs in Mated-1 h vs. Virgin-1 h group ((**a**) upregulated DEGs; (**b**) downregulated DEGs), Mated-6 h vs. Virgin-6 h group ((**c**) upregulated DEGs; (**d**) downregulated DEGs, no DEGs enriched), and Mated-24 h vs. Virgin-24 h group ((**e**) upregulated DEGs; (**f**) downregulated DEGs). The size of the dot indicates the number of DEGs in this pathway, and the color of the dot corresponds to different q-value ranges. The top 20 pathways each from upregulated DEGs and downregulated DEGs of each group are shown.

**Figure 6 insects-12-00693-f006:**
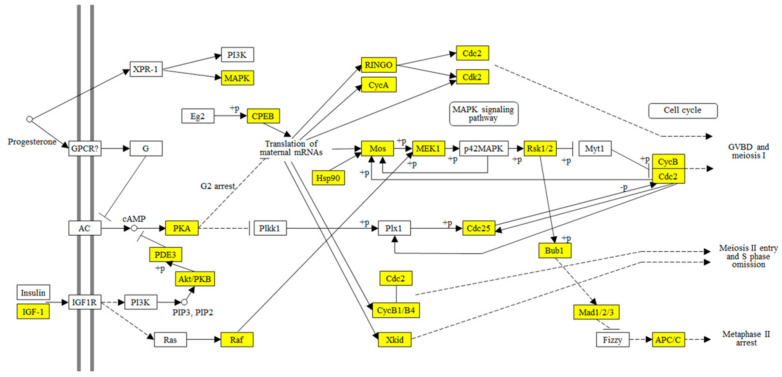
Mapping analysis of progesterone-mediated oocyte maturation pathway. Yellows represent the transcripts of *C. chuxiongica* females mapped to the Xenopus progesterone-mediated oocyte maturation pathway (KEGG id: map04914).

**Figure 7 insects-12-00693-f007:**
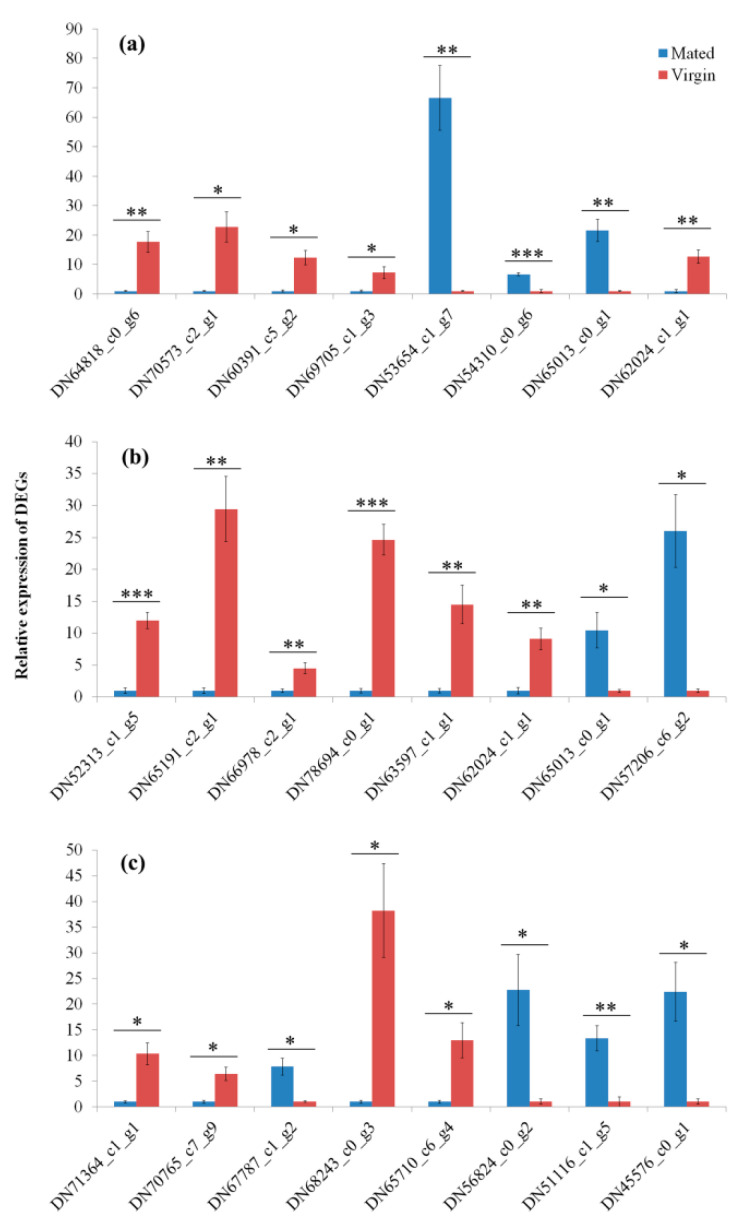
The validation of RNAseq results by quantitative real-time PCR. (**a**) DEGs from Mated-1 h vs. Virgin-1 h, (**b**) DEGs from Mated-6 h vs. Virgin-6 h, and (**c**) DEGs from Mated-24 h vs. Virgin-24 h. * indicates *p* < 0.05; ** indicates *p* < 0.01, *** indicates *p* < 0.001. Error bars indicate SE.

**Table 1 insects-12-00693-t001:** Mating-induced expression changes in genes related to reproduction in *C. chuxiongica* females.

GeneID	log2FoldChange	Padj	Annotation	Function	Reference
Mated-1 h vs. Virgin-1 h					
*DN64818_c0_g6*	−3.51325	0.02376	*Follicle cell protein 3C*	Encodes a major protein component of the vitelline membrane.	[33]
*DN64030_c0_g1*	9.66278	0.00015	*CYP2C (Cytochrome P450 family)*	Detoxification and the biosynthesis of hormones.	[34]
Mated-6 h vs. Virgin-6 h					
*DN65980_c5_g3*	−9.14964	1.05 × 10^−8^	*Vitellogenin-2*	Plays vital role in oocytes and embryo development in insects.	[35,36]
*N55390_c2_g3*	−7.75002	0.00289	*Vitellogenin-A2-like*		
*N52188_c2_g1*	−6.37534	0.00062	*Vitellogenin-A2-like*		
*DN71364_c1_g1*	−2.39804	2.43 × 10^−11^	*Vitellogenin-like*		
*DN59614_c4_g1*	−2.34561	1.55 × 10^−10^	*Vitellogenin-like*		
*DN78694_c0_g1*	−3.45314	0.03932	*Haemolymph juvenile hormone binding protein*	Protect the labile hormone molecules from degradation by esterase.	[37,38]
*DN63882_c0_g1*	−2.75932	0.00064			
*DN65663_c1_g1*	−2.48229	1.04 × 10^−8^			
*DN65191_c2_g1*	−2.34190	0.02528	*Protein takeout-like*	Takeout is part of a large gene family found throughout insects with roles in metabolism, circadian behavior, aging, and male courtship behavior.	[39]
*DN69612_c3_g1*	−1.58633	0.03611	*Protein takeout-like*		
*DN66978_c2_g1*	−2.71257	0.00840	*UDP-glucosyltransferase*	Positive modulator of fecundity.	[40]
*DN71428_c0_g1*	−8.16014	9.90 × 10^−5^	*Matrix metalloproteinase-16*	Some members of matrix metalloproteinase may play roles in corpus luteum formation, follicular development, and ovulation.	[41]
*DN69614_c4_g4*	−2.86011	0.04696	*CYP4 (Cytochrome P450 family)*	Detoxification and the biosynthesis of hormones.	[34]
*DN52313_c1_g5*	−2.81364	3.33 × 10^−14^	*CYP4 (Cytochrome P450 family)*		
*DN69106_c0_g1*	−2.15773	0.00170	*CYP4 (Cytochrome P450 family)*		
*DN53829_c6_g2*	−2.00810	0.01412	*CYP4 (Cytochrome P450 family)*		
Mated-24 h vs. Virgin-24 h					
*DN65980_c5_g3*	−9.44195	1.71 × 10^−14^	*Vitellogenin-2*	Plays vital role in oocytes and embryo development in insects.	[35,36]
*DN71047_c1_g1*	−8.99780	8.17 × 10^−11^	*Vitellogenin-A2-like*		
*DN55390_c2_g3*	−8.55112	1.39 × 10^−7^	*Vitellogenin-A2-like*		
*DN52188_c2_g1*	−8.51720	2.32 × 10^−7^	*Vitellogenin-A2-like*		
*DN49694_c0_g1*	−2.51669	0.00038	*Vitellogenin-like*		
*DN59614_c4_g1*	−2.30309	0.02439	*Vitellogenin-like*		
*DN71364_c1_g1*	−2.31618	0.02594	*Vitellogenin-like*		
*DN54912_c7_g1*	9.02029	1.51 × 10^−11^	*Vitellogenin receptor*		
*DN58744_c7_g1*	9.28253	8.58 × 10^−14^	*Zona pellucida sperm-binding protein (ZP) 1-like*		
*DN32813_c0_g1*	9.32544	3.72 × 10^−14^	*Zp1*		
*DN67043_c1_g1*	11.27687	2.07 × 10^−52^	*Zp1*		
*DN44634_c0_g2*	10.29005	2.97 × 10^−23^	*Zp2*		
*DN93387_c0_g1*	7.70158	0.002245	*Zp3*		
*DN50223_c1_g2*	8.04870	5.28 × 10^−5^	*Zp3*		
*DN12342_c0_g1*	8.89036	1.66 × 10^−10^	*Zp3*		
*DN63118_c0_g5*	9.43726	3.94 × 10^−15^	*Zp3*		
*DN42788_c0_g1*	9.81002	1.13 × 10^−18^	*Zp3*		
*DN44618_c0_g1*	9.87890	1.89 × 10^−52^	*Zp3*	The zona pellucida is a glycoprotein layer surrounding the plasma membrane of oocytes. It is a vital constitutive part of the oocyte and functions in primary binding and induction of the sperm acrosome reaction.	[42,43,44]
*DN43523_c0_g1*	10.0620	5.22 × 10^−21^	*Zp3*		
*DN22329_c0_g1*	10.18637	3.23 × 10^−22^	*Zp3*		
*DN47551_c0_g1*	10.48415	1.26 × 10^−43^	*Zp3*		
*DN48592_c0_g2*	11.49147	9.56 × 10^−37^	*Zp3*		
*DN63118_c0_g2*	11.49933	2.70 × 10^−64^	*Zp3*		
*DN47452_c0_g3*	11.88468	1.13 × 10^−41^	*Zp3*		
*DN49748_c0_g1*	12.50758	6.61 × 10^−49^	*Zp3*		
*DN100350_c0_g1*	12.60210	5.74 × 10^−50^	*Zp3*		
*DN63118_c0_g1*	13.09691	3.27 × 10^−70^	*Zp3*		
*DN67043_c2_g1*	7.96939	4.36 × 10^−26^	*Zp4*		
*DN67043_c1_g2*	8.81695	6.03 × 10^−10^	*Zp4*		
*DN72297_c0_g1*	9.81754	9.66 × 10^−19^	*Zp4*		
*DN67043_c2_g3*	10.63913	5.50 × 10^−62^	*Zp4*		
*DN67258_c0_g10*	10.78996	1.26 × 10^−28^	*Zp4*		
*DN50530_c5_g14*	9.970503	3.83 × 10^−20^	*ZP domain-containing protein-like*		
*DN46494_c0_g1*	12.88190	4.01 × 10^−53^	*ZP domain-containing protein-like*		
*DN64127_c3_g3*	9.64628	4.61 × 10^−17^	*Embryonic poly(A)-binding protein A*	Required for oocyte maturation and female fertility.	[45]
*DN82911_c0_g1*	8.20682	7.04 × 10^−6^	*Insulinase*	Destroys or inactivates insulin. Elevated insulin promotes oogenesis and inhibits immune responses.	[1,46]
*DN49876_c0_g1*	8.44020	2.62 × 10^−7^	*Insulinase*		
*DN46576_c0_g1*	8.18360	9.56 × 10^−6^	*G1/S-specific cyclin-E-like*	Oocyte maturation and the onset of embryogenesis.	[47,48,49]
*DN49363_c0_g1*	9.49947	1.08 × 10^−15^	*G2/mitotic-specific cyclin-A*		
*DN49147_c0_g1*	11.21366	1.90 × 10^−33^	*G2/mitotic-specific cyclin-B*		
*DN15801_c0_g1*	9.55009	3.72 × 10^−16^	*G2/mitotic-specific cyclin-B*		
*DN47388_c2_g1*	10.19335	2.76 × 10^−22^	*G2/mitotic-specific cyclin-B-like*		
*DN49534_c0_g1*	9.76583	2.58 × 10^−46^	*G2/mitotic-specific cyclin-B-like isoform X4*		
*DN8940_c0_g1*	10.33337	1.07 × 10^−23^	*Cell division cycle protein 20 homolog*		
*DN50119_c0_g2*	−6.06256	3.33 × 10^−13^	*Apolipophorin III*	Plays roles in lipid uptake by insect oocytes.	[50]
*DN71428_c0_g1*	−8.39452	1.39 × 10^−6^	*Matrix metalloproteinase-16*	Some members of matrix metalloproteinase may play roles in corpus luteum formation, follicular development, and ovulation.	[41]
*DN56705_c2_g7*	−3.79775	0.01004	*Matrix metalloproteinase-24-like*		
*DN37027_c0_g1*	11.90835	5.60 × 10^−42^	*Zygote arrest protein 1*	Ovary-specific maternal factor that plays essential roles during the oocyte-to-embryo transition.	[51]
*DN94405_c0_g1*	12.16383	4.47 × 10^−45^	*Zygote arrest protein 1*		
*DN84630_c0_g1*	8.95337	5.28 × 10^−11^	*Astacin-like; Ovastacin*	Cleaves ZP2 and prevents polyspermy; degrades egg envelope and releases the embryo from its intracapsular life.	[52]
*DN47329_c0_g1*	10.45829	5.41 × 10^−25^	*Oocyte-specific histone RNA stem-loop-binding protein 2*	Plays a role in the storage of replication-dependent histone mRNAs and proteins during oogenesis.	[53]

**Table 2 insects-12-00693-t002:** Reproductive related KEGG pathways of DEGs in *C. chuxiongica* females.

Pathway	Padj	Number of Genes	First Category	Second Category	Function
Mated-1 h vs. Virgin-1 h: Up regulated DEGs					
Steroid hormone biosynthesis	0.03602	1	Metabolism	Lipid metabolism	Egg maturation
Mated-24 h vs. Virgin-24 h: Down regulated DEGs					
Estrogen signaling pathway	0.01301	4	Organismal Systems	Endocrine system	Fecundity
Mated-24 h vs. Virgin-24 h: Up regulated DEGs					
Oocyte meiosis	0.01082	18	Cellular Processes	Cell growth and death	Egg maturation
Progesterone-mediated oocyte maturation	0.03177	13	Organismal Systems	Endocrine system	Egg maturation

**Table 3 insects-12-00693-t003:** Mating-induced expression changes in genes related to survivorship in *C. chuxiongica* females.

GeneID	log2FoldChange	Padj	Annotation	Function	Reference
Mated-1 h vs. Virgin-1 h					
*DN70573_c2_g1*	−5.42272	1.27 × 10^−9^	*Defensin*	Insect antimicrobial peptides.	[54]
*DN60391_c5_g2*	−5.02886	2.84 × 10^−8^	*Hymenoptaecin*		
Mated-6 h vs. Virgin-6 h					
*DN63597_c1_g1*	−3.53279	4.53 × 10^−12^	*Phenoloxidase*	Phenoloxidase involved in defensive melanization and production of oxidative free radicals.	[1]
Mated-24 h vs. Virgin-24 h					
*DN45974_c0_g1*	−6.77749	5.08 × 10^−7^	*Megourin*	Insect antimicrobial peptides.	[54]
*DN45254_c0_g1*	−6.34763	4.15 × 10^−17^	*Fungal protease inhibitor-1-like*	Against fungal infection.	[55]
*DN71273_c1_g5*	−2.95068	0.00896	*C-type lysozyme*	Defends against bacterial infection by hydrolyzing the bacterial cell walls and causing bacterial lysis.	[56]
*DN51648_c2_g8*	−6.63010	9.23 × 10^−11^	*Heat shock protein 27.2 (sHsp family)*	Heat shock and stress response.	[4]
*DN47336_c0_g1*	−5.01800	1.63 × 10^−12^	*Heat shock protein 27.2 (sHsp family)*		
*DN71108_c1_g3*	−4.29401	3.40 × 10^−11^	*Heat shock protein 27.2 (sHsp family)*		
*DN56298_c0_g1*	−3.69216	0.02212	*Heat shock protein 27.2 (sHsp family)*		
*DN70765_c7_g9*	−4.25919	1.24 × 10^−13^	*Heat shock protein 70-like (Hsp70 family)*		
*DN57953_c3_g2*	−3.39554	5.58 × 10^−10^	*Heat shock protein 70-like (Hsp70 family)*		
*DN62526_c5_g1*	−6.22937	1.06 × 10^−7^	*Heat shock protein 70 cognate (Hsp70 family)*		
*DN62526_c5_g4*	−4.59306	0.04497	*Heat shock protein 70 cognate (Hsp70 family)*		
*DN67613_c0_g10*	10.89725	7.42 × 10^−30^	*Heat shock protein 70 (Hsp70 family)*		
*DN67613_c0_g5*	8.479329	6.81 × 10^−21^	*Heat shock protein 70 (Hsp70 family)*		
*DN67090_c3_g8*	8.29444	2.51 × 10^−6^	*Heat shock protein 70 domain containing protein (Hsp70 family)*		
*DN64447_c3_g1*	7.39234	0.03894	*Heat shock protein 75 (Hsp90 family)*		
*DN64447_c4_g1*	3.92852	4.91 × 10^−11^	*Heat shock protein 90 (Hsp90 family)*		

**Table 4 insects-12-00693-t004:** Survivorship related KEGG pathways enriched in DEGs of *C. chuxiongica* females.

Pathway	Padj	Number of Genes	First Category	Second Category	Function
Mated-1 h vs. Virgin-1 h: Down regulated DEGs					
Inflammatory bowel disease (IBD)	0.02764	1	Human Diseases	Immune disease	Defense
Legionellosis	0.02981	1	Human Diseases	Infectious disease: bacterial	Defense
*Salmonella* infection	0.03185	1	Human Diseases	Infectious disease: bacterial	Defense
Pathogenic *Escherichia coli* infection	0.03341	1	Human Diseases	Infectious disease: bacterial	Defense
Mated-1 h vs. Virgin-1 h: Up regulated DEGs					
Inflammatory mediator regulation of TRP channels	0.03394	1	Organismal Systems	Sensory system	Defense
Complement and coagulation cascades	0.03397	1	Organismal Systems	Immune system	Defense
Mated-6 h vs. Virgin-6 h: Down regulated DEGs					
Complement and coagulation cascades	0.00061	5	Organismal Systems	Immune system	Defense
*Staphylococcus aureus* infection	0.00791	3	Human Diseases	Infectious disease: bacterial	Defense
Platelet activation	0.01418	4	Organismal Systems	Immune system	Defense
Mated-24 h vs. Virgin-24 h: Down regulated DEGs					
Antigen processing and presentation	0.00018	6	Organismal Systems	Immune system	Defense
Legionellosis	0.00046	6	Human Diseases	Infectious disease: bacterial	Defense
Measles	0.00319	5	Human Diseases	Infectious disease: viral	Defense
Complement and coagulation cascades	0.00620	4	Organismal Systems	Immune system	Defense
Influenza A	0.00987	5	Human Diseases	Infectious disease: viral	Defense
Toxoplasmosis	0.01130	4	Human Diseases	Infectious disease: parasitic	Defense
Tuberculosis	0.02795	4	Human Diseases	Infectious disease: bacterial	Defense
Amoebiasis	0.02986	3	Human Diseases	Infectious disease: parasitic	Defense
*Staphylococcus aureus* infection	0.04516	2	Human Diseases	Infectious disease: bacterial	Defense
Longevity regulating pathway - multiple species	0.01139	4	Organismal Systems	Aging	Aging
Mated-24 h vs. Virgin-24 h: Up regulated DEGs					
Pathogenic *Escherichia coli* infection	6.25 × 10^−5^	18	Human Diseases	Infectious disease: bacterial	Defense
Legionellosis	0.01178	16	Human Diseases	Infectious disease: bacterial	Defense
Viral myocarditis	0.02943	10	Human Diseases	Cardiovascular disease	Defense

## Data Availability

The transcriptome raw reads have been deposited to the NCBI SRA database (accession no.: SRR13991392~SRR13991407). Other data generated or analyzed during this study were included in this article and its Appendix A.

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
