# Peer review of "Mating-Induced Trade-Offs upon Egg Production versus Fertilization and Offspring’s Survival in a Sawfly with Facultative Parthenogenesis"

_insects, 2021, doi:10.3390/insects12080693_

Round 1
Reviewer 1 Report
Manuscript: Mating-induced trade-offs upon egg production versus fertilization and offspring’s survival in a sawfly with facultative parthenogenesis
General comments:
The manuscript reports on the clarification of important aspects of biology reproductive strategies of animals and and the relationships they establish with the environment, such trade-offs between reproduction and survival is conducive to provide evolutionary insights into reproductive strategies and aging for a serious pest of pines, Cephalcia chuxiongica Xiao (Hymenoptera: Pamphiliidae), in China and others countries. The topic addressed in the reproductive biology of insects is extremely important for the knowledge of the biology and evolution of the species and also for the applied scope.
The experiment was well designed, well executed and carefully analysed. In the manuscript, the objectives are well defined according to the experimental device presented. In addition to what has been achieved a very good framework literature on the subject and presented a novel approach methodology concerning the hormonal regulation expressed after mating.
For all these reasons, I recommend the publication; it only needs few corrections and add few missing data.
Specific comments:
Introduction
1) On line 96, replace C. by Cephalcia
Materials and Methods
1) On line 111, replace C. by Cephalcia
2) On line 125, clarify how it was guaranteed that the females were virgins
Results
No sugestions
Discussion
1) On line 385-386, the authors may provide some explanation for the existence of two mating systems, in particular the facultative parthenogenesis
References
Should the references not be presented in alphabetical order?
Reviewer 2 Report
The authors compared the general transcription levels after mating in a sawfly to infer the energy allocation strategy for reproduction, survival and immunity. Puzzling result is that vitellogenin related genes are down-regulated, that is counter-intuitive. ZP-related, insulin signals and HSP responses may be reasonable but the proof is lacking. The hypotheses established based on the genome reading must be confirmed in some way, eg. RNAi or al least 2D electrophoresis/MS analysis.
Reviewer 3 Report
The methods are sound, and the conclusions proceed well from the results. I am satisfied with the science behind the paper.
Multiple English errors need to be addressed by the editors, a native speaker, or a professional editing company.
Minor correction: lines 94-107 should not be the last paragraph of the introduction. End with a sentence like 91-93, explaining what you will do, and why this is valuable research.
Author Response
Reviewer 3#
Comments and Suggestions for Authors
The methods are sound, and the conclusions proceed well from the results. I am satisfied with the science behind the paper.
Multiple English errors need to be addressed by the editors, a native speaker, or a professional editing company.
Minor correction: lines 94-107 should not be the last paragraph of the introduction. End with a sentence like 91-93, explaining what you will do, and why this is valuable research.
Our answer:
We very appreciate the positive comments on our MS and the revision suggestions. We have now revised lines 94-107 accordingly (now please see lines 89-106).
We have had our MS edited by Dr. Su Ping Ong from Forest Research Institute Malaysia (FRIM) in the first round revision, which has been acknowledged in the Acknowledgement section. We have had this MS re-edited carefully by Dr. Su Ping Ong after the above revisions in this round. Changes due to English editing were highlighted by blue and other revisions according to reviewers’ comments were highlighted by red.
Submission Date
27 May 2021
Date of this review
16 Jul 2021 17:01:48

Round 2
Reviewer 2 Report
The authors failed to answer critical points I raised. Bioinformatic data themselves give us a landscape from that scientists can form a hypothesis. The hypothesis must be examined by experiments. That is the task of experimental biology. One of the easy tests is RNAi and we need this to conclude whether the hypothesis was right or wrong. This process is required and is not the subject of future study. Therefore I must consider the mission incomplete and the Editor should turn down its publication.
Author Response
|
Reviewer 2# round 2 |
Comments and Suggestions for Authors
The authors failed to answer critical points I raised. Bioinformatic data themselves give us a landscape from that scientists can form a hypothesis. The hypothesis must be examined by experiments. That is the task of experimental biology. One of the easy tests is RNAi and we need this to conclude whether the hypothesis was right or wrong. This process is required and is not the subject of future study. Therefore I must consider the mission incomplete and the Editor should turn down its publication.
Our answer:
We agree with the comments that the hypotheses established based on the genome reading can be confirmed by using RNAi and other techniques. We also feel sorry for not doing further experiments to archive full confirmation, which is due to short of funding and the two main authors graduated. However, we still believe that these results are worth publishing. Firstly, the findings and the hypotheses established in this study are informative and scientifically valuable for the knowledge of the biology and evolution of the species and also for the applied scope, which have also been confirmed by reviewers 1&3. Secondly, this study has well experimental design, extensive and in-depth data mining and analysis, reasonable discussion and careful presentation, which ensure the quality of this MS. Lastly, the number of related genes found in this study is large, the confirmation work of which should be huge; and thus publication the findings and established hypotheses at the current stage is helpful for other researchers to study this important species and its interesting biology and evolution.
Moreover, we have now added more discussion on this concern in the end of the last second paragraph in the Discussion section.
Submission Date
27 May 2021
Date of this review
29 Jun 2021 08:52:28
